# Learning to Customize Text-to-Image Diffusion In Diverse Context

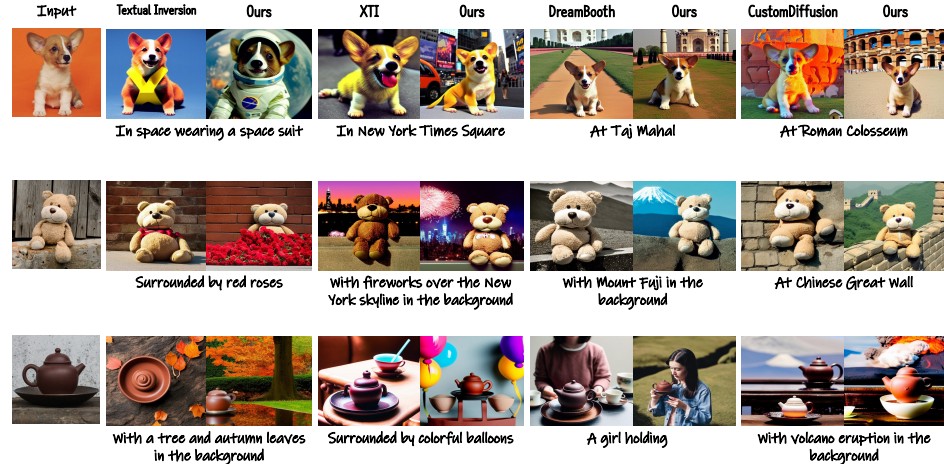

Figure 1: Comparison across various text-to-image models before and after integrating our method. The proposed approach consistently enhances prompt fidelity in generation results.

## Abstract

Most text-to-image customization techniques fine-tune models on a small set of *personal concept* images captured in minimal contexts. This often results in the model becoming overfitted to these training images and unable to generalize to new contexts in future text prompts. Existing customization methods are built on the success of effectively representing personal concepts as textual embeddings. Thus, in this work, we resort to diversifying the context of these personal concepts *solely* within the textual space by simply creating a contextually rich set of text prompts, together with a widely used self-supervised learning objective. Surprisingly, this straightforward and cost-effective method significantly improves semantic alignment in the textual space, and this effect further extends to the image space, resulting in higher prompt fidelity for generated images. Additionally, our approach does not require any architectural modifications, making it highly compatible with existing text-to-image customization methods. We demonstrate the broad applicability of our approach by combining it with four different baseline methods, achieving notable CLIP score improvements.

## 1 Introduction

Diffusion-based generative models (Ho et al., 2020; Song et al., 2020a; Dhariwal & Nichol, 2021; Song et al., 2020b) have made significant progress in image synthesis, achieving improved diversity and expressiveness in generated outputs. Extending these breakthroughs, diffusion-based text-to-image models (Rombach et al., 2022; Podell et al., 2023; Balaji et al., 2022; Saharia et al., 2022; Xue et al., 2024) that leverage large-scale text-image pairs (Schuhmann et al., 2021) have demonstrated impressive capabilities in translating the text into visual content.

More recently, leveraging the strong prior knowledge acquired by the pretrained text-to-image generative models, numerous approaches have been proposed to fine-tune the models for customization to specific concepts (Gal et al., 2022; Ruiz et al., 2023; Kumari et al., 2023; Voynov et al., 2023; Avrahami et al., 2023). Typically, these methods use 4-5 images containing personal concepts to obtain token embedding aligned with the given images, which are then integrated into the novel text

prompts for image generation. While demonstrating its potential, existing models often suffer from the *concept overfitting* when fine-tuned on a small set of images with limited contexts. This overfitting often causes the customized model to generate images that are highly similar to the training images, and fail to faithfully follow the text prompts during inference (Figure 1).

Our study indicates that introducing diverse contexts during model fine-tuning can mitigate the concept overfitting (Zeng et al., 2024). As diversifying text-image tuning pairs can be costly and often impractical, in this paper, we propose to diversify the context of personal concepts *solely* within the textual space, by first simply constructing a set of contextually diverse text prompts with concept tokens, nearly at no extra cost (Figure 2, left). As the customization aligns the personal concept with a concept token, this proposed approach can be a highly cost-effective way of context diversification. Then, we further adopt a self-supervised learning objective, Masked Language Modeling (MLM) (Devlin et al., 2018), which drives the concept embedding to learn proper relations to its contexts (Figure 2, right).

We later both theoretically and empirically show that adopting the MLM objective with a contextually diverse text prompt set during customization significantly alleviates the concept overfitting, and leads to semantic enhancement in textual representation, which ultimately extends to higher prompt fidelity in image generation. We conduct extensive experiments to demonstrate the effectiveness of our approach.

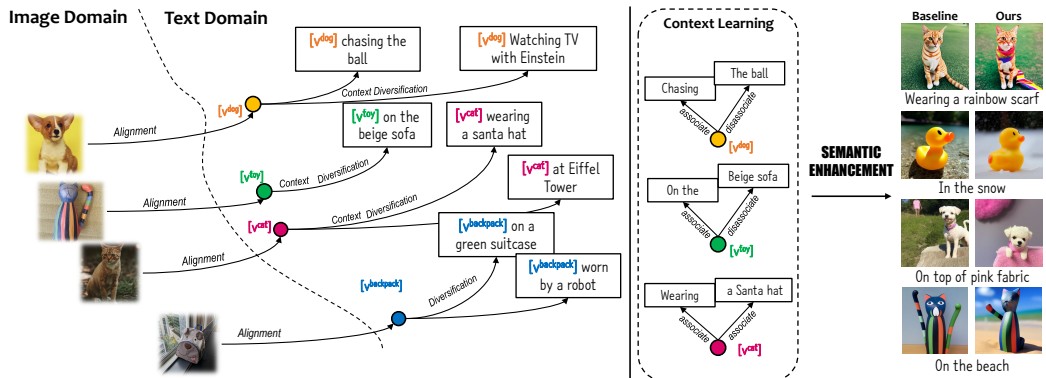

Figure 2: Conceptual illustration of the proposed approach. **Left:** We propose to diversify the context of the personal concept *solely* within the textual space, by simply constructing a context-rich text prompt set with a concept token. **Right:** In our method, the concept token embeddings are effectively guided to learn the relationship between the surrounding tokens in diverse contexts. This leads to the semantic enhancement of text representation by preserving the contextual information, which ultimately leads to higher text prompt fidelity in image generation. The proposed method is demonstrated both theoretically and empirically in the paper.

We summarize our contributions as follows,

- We propose a highly cost-effective text-to-image customization method that significantly improves context diversification of personal concepts via masked language modeling, leading to higher prompt fidelity in generated images.

- We theoretically illustrate that the proposed approach effectively helps to mitigate concept overfitting by regularizing the loss of contextual information and learning of diverse contexts.

- We further empirically show consistent image generation improvements while integrating our approach with four different text-to-image baseline methods, demonstrating its broad applicability.

## 2 RELATED WORKS

**Text-to-Image Generation.** The introduction of diffusion models (Ho et al., 2020; Nichol & Dhariwal, 2021; Song et al., 2020a) has paved the way for a series of text-to-image generative models (Saharia et al., 2022; Rombach et al., 2022; Balaji et al., 2022; Ramesh et al., 2021; 2022; Ding et al., 2022) that have achieved significant success. GLIDE (Nichol et al., 2021) demonstrated that using classifier-free guidance (Ho & Salimans, 2022) can enhance both the photorealism and caption alignment of generated images. DALLE-2 (Ramesh et al., 2022) further improved the process by leveraging CLIP (Radford et al., 2021) embeddings to derive an image prior from a text

caption, which is then decoded using diffusion models. Stable Diffusion (Rombach et al., 2022) proposed a way to improve the efficiency by applying the diffusion process in the lower dimensional latent space, and SDXL (Podell et al., 2023) has been proposed to make improvements over SD by updating the model architecture. ControlNet (Zhang et al., 2023) proposed to incorporate additional input conditions to improve the controllability of the T2I model using zero-convolutional layers. In our work, we mainly focus on baseline methods that are based on Stable Diffusion.

**Personalized Text-to-Image Genearation.** Textual inversion (Gal et al., 2022) pioneered a method to convert personal concept images into token embeddings, enabling the use of tokens for tailored text-to-image generation. DreamBooth (Ruiz et al., 2023) extended TI by fine-tuning the diffusion UNet along with the prior-preservation loss to prevent forgetting of prior concepts. Since the introduction of the pioneering works, a line of work has been proposed to make further improvements. XTI (Voynov et al., 2023) proposed to invert the concept into multiple token embeddings, each specialized for a different layer of the diffusion network. CustomDiffusion (Kumari et al., 2023) proposed to fine-tune the cross-attention layer in diffusion UNet for efficient training. Break-A-Scene (Avrahami et al., 2023) proposed to learn multiple concepts included in the same scene by utilizing masked diffusion loss. There is also a growing interest in developing methods specialized for facial images. (Yuan et al., 2023) constructed a set of basis tokens corresponding to celebrities and optimized their weights to synthesize a given image. (Peng et al., 2024) proposed to augment the concept token embedding by extracting facial features using a facial recognition model. (Shi et al., 2024) have proposed a test-time finetuning-free method where a learnable image encoder is deployed to convert the input images into a textual token. (Chen et al., 2024) also proposed an instant method where an apprentice diffusion model learns to imitate the behaviors of multiple expert models specialized for each concept. Similarly, (Wei et al., 2023) proposed to use a CLIP image encoder to encode personal images and then utilize global and local mapping networks to obtain enhanced representations of the concept. (Chen et al., 2023) proposed a method to avoid the entanglement of identity-irrelevant features by utilizing learnable masks and multi-task training objectives.

## 3 PRELIMINARIES

### 3.1 TEXT-TO-IMAGE GENERATION

We apply our approach to various text-to-image baseline models (Gal et al., 2022; Ruiz et al., 2023; Voynov et al., 2023; Kumari et al., 2023) that are based on Stable Diffusion (SD) (Rombach et al., 2022). SD consists of a CLIP text encoder $\Gamma$ that encodes an input text $\mathbf{t}$ into a sequence of input token embeddings, denoted as Tokenize, $\mathbf{P} = \text{Tokenize}(\mathbf{t})$, then outputs corresponding text embedding $\mathbf{C} = \Gamma(\mathbf{P})$ using self-attention layers. A Variational Auto Encoder (VAE) of SD $\mathcal{E}$ encodes an image $x$ to a lower dimensional latent $\mathbf{z} = \mathcal{E}(\mathbf{x})$.

During training, given a timestep $t \sim \text{Uniform}[0, \text{T} - 1]$, a random noise map $\epsilon \sim \mathcal{N}(\mathbf{0}, \mathbf{I})$ is added to the latent map to get a noised latent map $\mathbf{z}_t = \alpha_t \mathbf{z} + \sigma_t \epsilon$. Then, the diffusion U-Net $\epsilon_\theta$ is trained to minimize the following objective for denoising,

$$\mathbb{E}_{\mathbf{C}, \epsilon, t, \mathbf{z}} ||\epsilon - \epsilon_\theta(\mathbf{z}_t, t, \mathbf{C})||_2^2. \tag{1}$$

### 3.2 FINETUNING FOR TEXT-TO-IMAGE CUSTOMIZATION

Utilizing a text prompt $\widetilde{\mathbf{t}}$ that incorporates a concept token $*$ (e.g., "a picture of a [*] dog"), the tokenized input embedding is encoded $\widetilde{\mathbf{C}} = \Gamma(\widetilde{\mathbf{P}})$. Following the text encoding, the denoising objective is computed as below,

$$\mathcal{L}_{\text{Custom}}(\mathbf{z}_t, t, \widetilde{\mathbf{C}}) := \mathbb{E}_{\widetilde{\mathbf{C}}, \epsilon, t, \mathbf{z}} ||\epsilon - \epsilon_\theta(\mathbf{z}_t, t, \widetilde{\mathbf{C}})||_2^2, \tag{2}$$

where $\mathcal{L}_{\text{Custom}}$ denotes the denoising loss utilized for model customization, with $\mathbf{z} = \mathcal{E}(\mathbf{x})$ encoding an image $\mathbf{x}$ sampled from a small set of personal concept images. Depending on the baseline method, a different set of parameters are optimized. For Textual Inversion (TI) (Gal et al., 2022), only the concept token embedding is optimized with respect to Eqn. 2. Our method does not require any architectural modification, hence it is highly compatible with existing methods. We demonstrate this applicability by applying our method to different baselines. Unless otherwise specified, we illustrate our method using TI.

## 4 METHOD

Text-to-image customization methods (Gal et al., 2022; Voynov et al., 2023; Ruiz et al., 2023; Kumari et al., 2023), typically trained on 4-5 images with limited context, are prone to be overfitted to

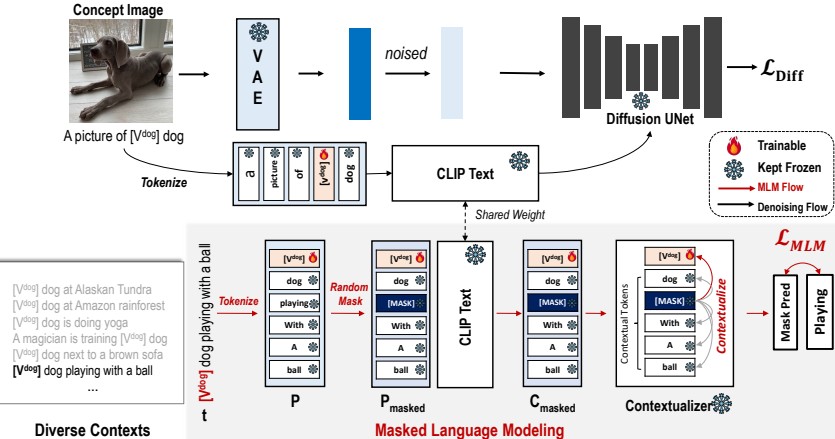

Figure 3: Illustration of the proposed text-to-image customization process. The MLM loss $\mathcal{L}_{\text{MLM}}$ is computed, along with the denoising loss $\mathcal{L}_{\text{Diff}}$, to align the special concept image with the concept token, For MLM, we sample text prompts from a contextually diverse prompt set. The sampled prompt is then tokenized and mapped to a prompt embedding $\mathbf{P}$. Subsequently, a subset of the input tokens are masked to yield $\mathbf{P}_{\text{masked}}$, and fed into CLIP text encoder to output $\mathbf{C}_{\text{masked}}$. Then, the masked embedding is contextualized with the surrounding tokens, including the concept token and the context tokens, by self-attention layers. After that, the masked token is predicted. As the concept token is trained to predict the best semantically aligned token with $\mathcal{L}_{\text{MLM}}$, the concept token embedding effectively learns its context. For computing $\mathcal{L}_{\text{Diff}}$, we use the context-simple caption, the same as the baseline. Textual Inversion (Gal et al., 2022) is used as an example baseline here.

the training set. To address this issue, we propose contextual diversification *solely* within the textual space by constructing a context-rich text prompt set. To effectively guide the concept embedding to learn the proper contextual semantics, we adopt masked language modeling (MLM) during customization (Section 4.1 and 4.2), which leads to semantic enhancement in both textual (4.3) and image space (Section 4.4).

## 4.1 MASKED LANGUAGE MODELING

In order to enhance the *concept token* embedding with context-rich text prompts, we adopt Masked Language Modeling (MLM) during the model customization. The overall process is illustrated in Figure 3. We elaborate on the corresponding details in the following steps,

(i) A text prompt $\mathbf{t}$, drawn from a contextually diverse prompt set that includes the *concept* token, e.g., [V$^{\text{dog}}$], is tokenized and mapped to a prompt embedding,

$$\mathbf{P} = \text{Tokenize}(\mathbf{t}), \tag{3}$$

where $\mathbf{P} \in \mathbb{R}^{L \times d}$, $L$ is the number of tokens, $d$ is the feature dimension of prompt embeddings.

(ii) A subset of prompt embedding $\mathbf{P}$ is randomly selected and those selected tokens are replaced by a mask token embedding $p_{\text{mask}}$ with the probability $\rho_{\text{mask}}$, yielding $\mathbf{P}_{\text{masked}} = \text{RandomMask}(\mathbf{P}, \rho_{\text{mask}})$. Then, text embedding is obtained from CLIP text encoder $\Gamma$,

$$\mathbf{C}_{\text{masked}} = \Gamma(\mathbf{P}_{\text{masked}}), \tag{4}$$

where $\mathbf{C}_{\text{masked}} = \{c_i\}$, with $c_i \in \mathbb{R}^d$ denoting one element in token embedding.

(iii) Finally, we predict the label of the masked token $\hat{\mathbf{y}} = \psi(\mathbf{C}_{\text{masked}})$ and calculate the MLM loss,

$$\mathcal{L}_{\text{MLM}} = \mathbb{E}\Big[\text{CrossEntropy}(\mathbf{y}, \hat{\mathbf{y}})\Big], \tag{5}$$

where $\psi$ denotes the classification network with self-attention layers.

Notably, the attention layer computes the output token embedding $\mathbf{O}$ as the linear combination of input embeddings, with the weights determined by the self-attention map $\mathbf{A}^{\text{self}} \in \mathbb{R}^{L \times L}$, implying that the output is the *contextualization* of the input. For the $i_m$-th output token, i.e., the masked

token, it can be formulated as below,

$$\underbrace{\mathbf{O}[i_m,:]}_{\text{Output Mask}} = \sum_{j=1}^{L} \mathbf{A}^{\text{self}}[i_m,j]\mathbf{V}[j,:] = \sum_{\substack{j=1 \\ j \neq j_*}}^{L} \mathbf{A}^{\text{self}}[i_m,j]\underbrace{\mathbf{V}[j,:]}_{\text{Context}} + \mathbf{A}^{\text{self}}[i_m,j_*]\underbrace{\mathbf{V}[j_*,:]}_{\text{Concept}}, \tag{6}$$

where $j_*$ is the index of the concept token, and $\mathbf{V}$ is the value matrix.

By optimizing the concept token embedding to minimize $\mathcal{L}_{\text{MLM}}$, the concept token is guided to learn the diverse context, as the MLM encourages the utilization of surrounding contexts for mask prediction. The overall process is illustrated in Figure 3. We later show that customization with this additional objective results in regularizing the text embeddings from overfitting to the concept token, which eventually leads to semantically enhanced image generation (Section 4.4).

---

**Algorithm 1** Training Procedure of Contextualizer

---

1: Load parameters $\Gamma$                                                                 $\{\Gamma: \text{CLIP text}\}$
2: Random initialize $\psi, p_{\text{mask}}$                          $\{\psi: \text{Contextualizer}, p_{\text{mask}}: \text{mask embedding}\}$
3: Set $\rho_{\text{mask}}$                                                         $\{\rho_{\text{mask}}: \text{masking probability}\}$
4: **repeat**
5:     Sample $\mathbf{t}$ from rich prompt set, $\mathbf{P}=\text{Tokenize}(\mathbf{t})$
6:     $\mathbf{P}_{\text{masked}}, \mathbf{y} = \text{RandomMask}(\mathbf{P}, \rho_{\text{mask}})$                         $\{\mathbf{y}: \text{masked token label}\}$
7:     Compute $\mathcal{L}_{\text{MLM}} = \mathbb{E}_{\mathbf{y}, \mathbf{P}_{\text{masked}}}\Big[\text{CrossEntropy}((\mathbf{y}, \psi(\Gamma(\mathbf{P}_{\text{masked}}))))\Big]$
8:     Gradient descent optimization on $\nabla_{\psi, p_{\text{mask}}}\mathcal{L}_{\text{MLM}}$
9: **until** optimized

---

## 4.2 Customization with Diverse Context

**Prompt Set Construction.** To achieve customization with diverse contexts, we construct a set of context-rich prompts that incorporate the special concept token. Inspired by recent work (Brooks et al., 2023), we leverage a pretrained large language model (OpenAI, 2023; Brown et al., 2020) to minimize the effort in manually crafting a large set of prompts. For this, we query the LLM to generate a list of contexts of different types, e.g., background or subjection variation. For detailed descriptions of the prompt set construction process, refer to the Appendix Section A.2.

**Pretraining.** Although the CLIP text encoder of SD has the linguistic capability of comprehending text prompts, it is solely trained with contrastive learning objectives (Radford et al., 2021), and does not support MLM. Therefore, before proceeding with fine-tuning for customization, we first pretrain a network, namely, a *contextualizer* $\psi$ to incorporate the MLM capability. We provide the pretraining procedure of the contextualizer $\psi$ in Algorithm 1. During the pretraining of $\psi$, the concept token is not involved. We only train the mask embedding and the layers of the contextualizer. The CLIP text encoder and diffusion U-Net remain fixed.

**Finetuning.** Utilizing the contextually diverse prompt set, we proceed with the model customization (Figure 3). The model optimized by minimizing the denoising objective $\mathcal{L}_{\text{Diff}}$ (Eqn. 2) and the MLM loss $\mathcal{L}_{\text{MLM}}$ (Eqn. 5). Two different types of prompts are utilized for each objective. Text embeddings $\widetilde{\mathbf{C}}$ encoded from a context-simple text prompt $\widetilde{\mathbf{t}}$ (e.g., *"a picture of [v] dog"*) for $\mathcal{L}_{\text{Diff}}$, and text embeddings $\mathbf{C}_{\text{masked}}$ encoded from a text context-rich text prompt $\mathbf{t}$ (e.g., *"a [v] dog at Eiffel Tower"*) for $\mathcal{L}_{\text{MLM}}$. The overall learning objective can be formulated as follows,

$$\mathcal{L}_{\text{Diff}}(\mathbf{z}_t, t, \widetilde{\mathbf{C}}) + \lambda \mathcal{L}_{\text{MLM}}(\mathbf{C}_{\text{masked}}), \tag{7}$$

where $\mathbf{z}$ denotes the noised latent of personal concept image, $t$ denotes the timestep, and $\lambda$ denotes the weight for the MLM loss. Note that, as the MLM objective does not require *any* images, the concept token embedding learns contextual semantics without constructing corresponding images. Additionally, our approach does not require *any* architectural modification of SD, it is highly compatible with existing text-to-image approaches. Hence, we combine our approach with different baseline methods and demonstrate its generalizability. The overall training procedure is described in Algorithm 2. We refer to Appendix Section A.1 for additional details of training.

## 4.3 Semantic Enhancement in Textual Space

In this section, we illustrate how adopting the MLM with diverse contexts during the model customization leads to semantically enhanced textual representation, which ultimately translates to improved image generation. We first validate the following to explain how our method leads to semantically enhanced textual representation,

---

**Algorithm 2** Training Procedure of Text-to-Image Customization

---

1: Load parameters $\theta$, $\psi$, $\Gamma$, and $\mathcal{E}$         $\{\theta$: U-Net, $\psi$: contextualizer, $\Gamma$: CLIP text, $\mathcal{E}$: VAE$\}$
2: Fix $\psi$ and $p_m$         $\{p_m$: mask embedding$\}$
3: Set $\lambda$ and $\rho_{\text{mask}}$         $\{\rho_{\text{mask}}$: masking probability$\}$
4: Select trainable params $\Theta \subset \{\theta, \Gamma\}$         $\{$Selection based on baselines$\}$
5: **repeat**
6:      Sample $t \sim \text{Uniform}[0, \text{T} - 1]$, $\epsilon \sim \mathcal{N}(\mathbf{0}, \mathbf{I})$
7:      Sample $\mathbf{x}, \mathbf{P}$ and encode $\mathbf{z} = \mathcal{E}(\mathbf{x})$, $\mathbf{c} = \Gamma(\mathbf{P})$
8:      Get noised latent, $\mathbf{z}_t = \alpha_t \mathbf{z} + \sigma_t \epsilon$
9:      $\mathbf{P}_{\text{masked}}, \mathbf{y} = \text{RandomMask}(\mathbf{P}, \rho_{\text{mask}})$         $\{\mathbf{y}$: masked token label$\}$
10:      Compute $\mathcal{L}_{\text{Diff}} = \mathbb{E}_{\mathbf{C}, \epsilon, t, \mathbf{z}} ||\epsilon - \epsilon_\theta(\mathbf{z}_t, t, \mathbf{C})||_2^2$
11:      Compute $\mathcal{L}_{\text{MLM}} = \mathbb{E}_{\mathbf{y}, \mathbf{P}_{\text{masked}}} \Big[ \text{CrossEntropy}(\mathbf{y}, \Gamma(\mathbf{P}_{\text{masked}})) \Big]$
12:      Gradient descent optimization on $\nabla_\Theta \Big[ \mathcal{L}_{\text{Diff}} + \lambda \mathcal{L}_{\text{MLM}} \Big]$
13: **until** optimized

---

- The model *overfits* to the personal concept, when the semantics of the *context* tokens (*i.e.,* non-personal) become *similar* to the *concept* token.
- The semantics of the *context* tokens get *distinct* from the *concept* token as the diverse contextual semantics are learned with MLM.

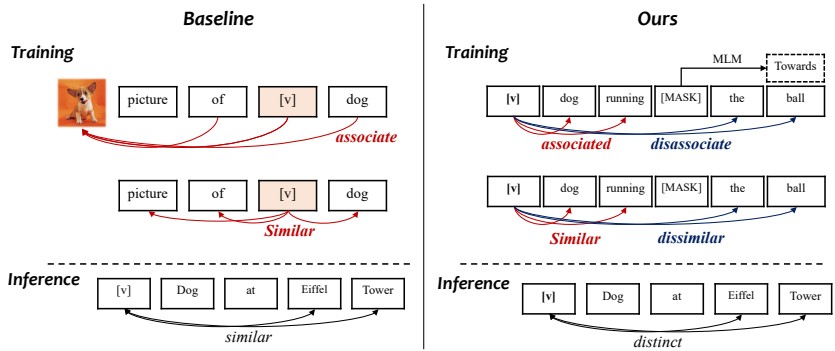

Figure 4: Illustrative comparison between the baseline approach and ours. **Left:** The baseline approach is prone to losing the semantics of the contexts, as the concept token embedding *only* learns to associate the tokens within limited contexts that correspond to the same concept image. As a result, the semantics of the distinct subject tokens become similar, leading to *concept overfitting*. **Right:** In contrast, MLM regularizes the loss of contextual semantics, as their elimination leads to ineffective mask predictions. Also, by deploying MLM with diverse contexts, the concept token embedding learns to *both* associate and disassociate the context tokens. By learning to disassociate the distinct subject, the contextual semantics are preserved.

Most customization methods that fine-tune the model with limited context (Gal et al., 2022; Ruiz et al., 2023; Voynov et al., 2023; Kumari et al., 2023) often suffer from *concept overfitting* (Zeng et al., 2024), resulting in generated images that primarily contain the personal concept without adhering to the prompt. We next analyze that concept overfitting leads to a high similarity between the text embeddings of *context* tokens and the *concept* token, ultimately causing a loss of contextual semantics.

**Proposition 1.** *The model overfitting to the concept token makes the attention map mostly attend to the concept token, i.e., $A[i, j_*] \gg A[i, j], \forall j \neq j_*$, where $j_*$ is the index of the concept token. The distance between the context embeddings $c_i$ and the concept embedding $c_{i_*}$ is bounded,*

$$||c_i - c_{i_*}||_2 \leq \delta_V. \tag{8}$$

In contrast, the MLM regularizes the loss of contextual information and guides the learning of diverse contexts (Figure 4, right). Specifically, we focus on two types of text embeddings for the concept token: (i) the embedding derived from the prompt *with* the concept token, referred to as $c_b$, which represents the embeddings obtained through a typical customization method; and (ii) the

embedding from the prompt *without* the concept token, denoted as $\hat{c}_b$, which are the proposed embeddings with a low MLM loss, i.e., the desired embeddings we aim to achieve.

**Proposition 2.** *Optimizing the concept token $p_*$ with the MLM loss $\mathcal{L}_{MLM}$, the minimized distance between the text embedding of context token $c_b$ and $\hat{c}_b$ is the necessary condition to minimize $\mathcal{L}_{MLM}(c_b)$, i.e.,*

$$\mathcal{L}_{MLM}(c_b) - \mathcal{L}_{MLM}(\hat{c}_b) \leq \delta_g ||c_b - \hat{c}_b||_2. \tag{9}$$

**Remark 3.** *According to Proposition 1, solely optimizing the concept token tends to produce text embeddings of context token $c_b$ that closely resemble the text embedding of concept token $c_*$ but deviate from desired embeddings $\hat{c}_b$. However, as outlined in Proposition 2, incorporating MLM can significantly align $c_b$ with $\hat{c}_b$.*

We provide the proof of Proposition 1 and 2 in Appendix A.4.

To empirically validate this, we provide a cosine similarity analysis using a set of 200 text prompts with concept tokens and varying context tokens (Table 1). We analyze the cosine similarity, $sim_1 = cos(c_{i_*}, c_b)$, and $sim_2 = cos(c_b, \hat{c}_b)$, respectively. The result shows that the baseline approach leads to high similarity between the concept-context tokens ($sim_1$) and low similarity in the context token from the two prompts ($sim_2$). The baseline result implies that the contextual semantics are not only getting similar to the concept (high $sim_1$) but also implies that the context token loses its semantics as it becomes dissimilar to the semantics that are preserved in the context token (low $sim_2$). In contrast, we observe the opposite trend (low $sim_1$ and high $sim_2$), which implies that the MLM mitigates the concept overfitting and encourages the contextual semantics to be distinct.

| Method | $sim_1 \downarrow$ | $sim_2 \uparrow$ | $SKL \uparrow$ |
|---|---|---|---|
| Baseline | 0.5047 | 0.3864 | 1.5932 |
| Ours | **0.4072** | **0.7386** | **2.3536** |

Table 1: Cosine similarity between the concept and context token from the same prompt ($sim_1$) and the context tokens from different prompts $sim_2$ are reported. $SKL$ denotes the symmetric KL divergence between the cross attention maps of the concept token and the context token.

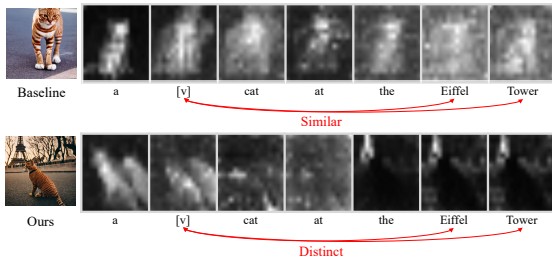

Figure 5: Visualization of $16 \times 16$ attention maps from cross-attention layers. **Top:** Baseline. **Bottom.** Our approach results in cross-attention maps of the concept token and the context token being more distinctively distributed, leading to semantically enhanced image generation.

### 4.4 SEMANTIC ENHANCEMENT IN IMAGE SPACE

The cross-attention map plays a key role in controlling the overall image generation (Hertz et al., 2022; Chefer et al., 2023), where the attended region corresponds to the area most influenced by the token. Next, we provide an analysis of the cross-attention map to illustrate how the aforementioned semantic enhancement in textual space can be transferred to image space, thereby improving the prompt fidelity of image generation.

Let $\mathbf{Q}_{\mathcal{I}}$, $\mathbf{K}_{\mathcal{T}}$ and $\mathbf{V}_{\mathcal{T}}$ denote the Query, Key and Value of the cross attention layer, projected from image $\mathcal{I}$, and text $\mathcal{T}$. During the denoising process, the cross attention map $\mathbf{A}^{\text{cross}} = \text{Softmax}(\frac{\mathbf{Q}_{\mathcal{I}}\mathbf{K}_{\mathcal{T}}^{\top}}{\sqrt{d}})$ is computed between $\mathbf{Q}_{\mathcal{I}} \in \mathbb{R}^{|queries| \times d}$ and $\mathbf{K}_{\mathcal{T}} \in \mathbb{R}^{L \times d}$, where $|queries|$ denotes the number of image tokens, $L$ denotes the number of text tokens, and $d$ denotes the dimension of each image/text token embedding.

**Proposition 4.** *Denote the correlation between the text embedding $c_i$ and image embeddings as $\boldsymbol{M}[:, i] = \boldsymbol{Q}_{\mathcal{I}}\boldsymbol{K}_{\mathcal{T}}[i, :]$. $\boldsymbol{M}[:, i]$ and $\boldsymbol{M}[:, j]$ are bounded by the distance between their corresponding text embeddings $c_i$ and $c_j$,*

$$||\boldsymbol{M}[:, i] - \boldsymbol{M}[:, j]||_2 \leq \alpha ||c_i - c_j||_2, \tag{10}$$

*where $\alpha = ||\boldsymbol{Q}_{\mathcal{I}}||_F ||\mathbf{W}_K||_F$.*

**Remark 5.** *For the baseline method, Proposition 1 shows that the distance between text embeddings of context tokens $c_b$ and concept token $c_*$ are bounded by a small value, which means $c_b$ and $c_*$ are similar. Furthermore, Proposition 4 suggests the image embedding corresponding to $c_b$ highly resembles the one corresponding to $c_*$. This suggests that the generated images are likely to focus solely on the object of the personal concept, overlooking the context.*

Table 2: Evaluation results of the proposed approach combined with baselines. The winning result between the baseline and ours is denoted in **bold**.

| | TI | | XTI | | DB | | CD | |
|---|---|---|---|---|---|---|---|---|
| | **Baseline** | **Ours** | **Baseline** | **Ours** | **Baseline** | **Ours** | **Baseline** | **Ours** |
| **CLIP-T↑** | 0.279 | **0.305** | 0.297 | **0.305** | 0.306 | **0.309** | 0.322 | **0.326** |
| **DINO↑** | **0.556** | 0.543 | 0.586 | **0.594** | **0.667** | 0.655 | **0.618** | 0.615 |
| **DINO-FG↑** | 0.661 | **0.658** | 0.692 | **0.710** | **0.783** | 0.775 | **0.737** | 0.736 |

**Remark 6.** *For our method, Proposition 2 shows that the distance between the text embeddings of concept token $c_b$ and the text embeddings of desired concept token $\hat{c}_b$ should be small. The image embedding corresponding to $c_b$ should highly resemble the one corresponding to $\hat{c}_b$. This indicates that the contexts in the generated images will remain consistent with their text prompts.*

We provide the proof of Proposition 4 in Appendix A.5.

We visualize the cross-attention maps of the tokens and compare the results of the baseline (Gal et al., 2022) and ours to further validate our claims (Figure 5). The result indicates that concept overfitting of the text embedding in the baseline approach leads to cross-attention maps of the concept token and the context token being more closely distributed, leading to semantically degraded image generation. Using a prompt set of size 200 containing both context token and concept token, we measure the symmetric KL divergence between the cross-attention maps of the concept tokens $c_*$ and the context tokens $c_b$ within the same prompt: $SKL = \frac{1}{2}D_{\text{KL}}(\mathbf{A}^{\text{cross}}[:, *]||\mathbf{A}^{\text{cross}}[:, b]) + \frac{1}{2}D_{\text{KL}}(\mathbf{A}^{\text{cross}}[:, b]||\mathbf{A}^{\text{cross}}[:, *])$, where $D_{\text{KL}}$ is the Kullback-Leibler (KL) divergence, $\mathbf{A}^{\text{cross}}[:, k] \in \mathbb{R}^{|queries|}$ denotes the cross attention map of the $k$-th token. The results show that the baseline approach produces a more similar distribution of the cross-attention maps between the concept and context tokens (Table 1, $SKL$). This finding, along with our visual evidence (Figure 4), indicates that semantic enhancement in textual space leads to enhancement in image space, resulting in improved prompt fidelity of the image generation.

## 5 EXPERIMENTS

### 5.1 EXPERIMENTAL SETTINGS

**Baselines.** We apply our approach to four different baselines. Apart from the adoption of the MLM objective during model customization, the remaining training configuration remains the same for the baselines and ours. **Textual Inversion (TI)** (Gal et al., 2022), **XTI** (Voynov et al., 2023), **Dreambooth (DB)** (Ruiz et al., 2023) and **CustomDiffusion (CD)**. For each baseline, the training parameters are chosen following the original configuration. For TI and XTI, only the personal concept embeddings are updated. For DB, we finetune the entire parameters of the U-Net and the CLIP text encoder. For CD, we train the personal concept embedding and the Key/Value projection matrices of cross-attention layers of the U-Net. For all the prompts, we use a joint phrase that combines the special token with the prior concept (e.g., '[v] dog'). We do not mask the special tokens, and we set the $\rho_{\text{mask}} = 15\%$ following (Devlin et al., 2018). we use AdamW (Loshchilov, 2017) optimizer to update the parameters on a single NVIDIA RTX 3090 GPU. We provide additional implementation details of the baseline methods in Appendix Section A.1.

Table 3: Evaluation results with varying $\lambda$. The best results are denoted in **bold**.

| $\lambda$ | CLIP-T↑ | DINO-FG↑ |
|---|---|---|
| Baseline | 0.279 | 0.657 |
| 0.00001 | 0.292 | **0.659** |
| 0.0001 | 0.305 | 0.658 |
| 0.0005 | **0.311** | 0.654 |

Table 4: Ablation study on masking probability. The best results are denoted in **bold**.

| $\rho_{\text{mask}}\%$ | CLIP-T↑ |
|---|---|
| Baseline | 0.279 |
| 15 | **0.305** |
| 50 | 0.304 |
| 90 | 0.299 |

**Dataset.** We use a mixture of 15 different subjects adopted from DB (Ruiz et al., 2023), TI (Gal et al., 2022) and CD (Kumari et al., 2023). We use 11 subjects from DB (Ruiz et al., 2023) composed of: [backpack,backpack_dog, cat, cat2, cat3, cat3, cat6, duck_toy, poop_emoji, rc_car, teapot, teddybear], and we use 3 subjects from (Kumari et al., 2023): [pet_cat1, pet_dog1, wooden_pot]. Finally, we use 1 subject from (Gal et al., 2022): [cat toy]. For each subject, utilize 4 to 5 images are provided. For benchmark prompts, a benchmark prompt set from DB is utilized. This prompt set contains 25 prompts, and we generate 8 images per prompt. In total, 3,000 images are generated for each experiment.

**Evaluation Metrics.** Following the literature (Ruiz et al., 2023; Gal et al., 2022; Kumari et al., 2023) we first measure the text prompt fidelity. For this measure, the average pairwise cosine similarity between the input prompt and the generated image encoded by the CLIP text/vision encoder is used (**CLIP-T**). We also measure the subject fidelity. For this measure, the average pairwise cosine similarity between the personal concept image and the generated image encoded by the ViT-S/16 DINO (Caron et al., 2021) embeddings is used (**DINO**). Following Kim et al. (2024), we measure the subject fidelity by computing DINO (Caron et al., 2021) score on the segmented foreground region (**DINO-FG**). We obtain segmentation masks with Grounded-SAM (Ren et al., 2024) conditioned on the subject's class name. As it removes the influence of the background, this leads to more accurate measuring of subject fidelity.

## 5.2 QUANTITATIVE RESULTS

**Comparison with Baseline Method.** We combine the proposed approach with four different baselines and present the quantitative comparisons (Table 2). Notably, for all baseline methods, we achieve consistent improvement in semantic alignment between the generated images and the input prompts, as we observed improvement in **CLIP-T** score for all baselines. Compared to the methods that update the parameters other than the personal token embeddings (DB and CD), the ones that do not update them show higher improvement. We hypothesize that as the model that trains U-Net still utilizes the contextually limited text-image pairs for the denoising objective, this leads to overfitting of the cross-attention layers. As a result, the enhancement property of our method can not be faithfully transferred to the image space. For the subject fidelity measure, we observe a difference from the baseline methods. We later show that prompt-subject fidelity trade-off can be achieved by different $\lambda$ (Section 5.3).

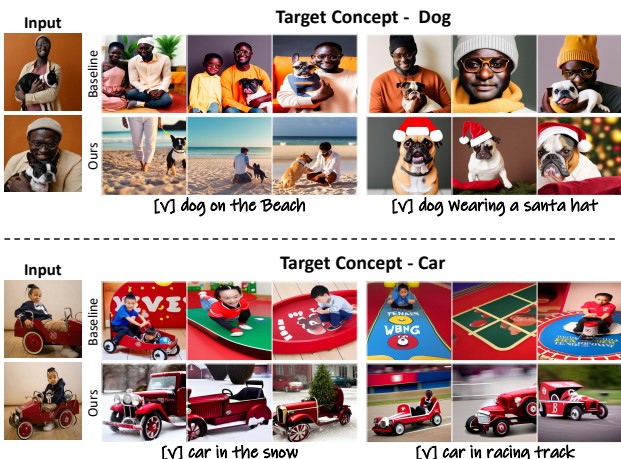

Figure 6: Customization on multi-concept images. Along with the simple text prompt (e.g., *"a picture of [v] dog"*) for the denoising objective $\mathcal{L}_{\text{Diff}}$, we construct a set of prompts tailored to the concept for the MLM objective $\mathcal{L}_{\text{MLM}}$ (e.g., *"a man is petting a [v] dog"*). The result clearly indicates that our method drives the concept embedding to focus on the target concept.

## 5.3 ABLATION STUDY

In this section, we conduct ablation studies to provide deeper insights into our method and validate its effectiveness. To solely compare the effect of adjusting the textual space, we have UNet and CLIP fixed and only update the concept token embedding.

**Impact of $\lambda$.** We first study the impact of MLM objective weight (Table 3). We adopt TI as our baseline and compare the prompt and subject fidelity with the model trained with different $\lambda$. We note the general trend of improved **CLIP-T** score as the model is trained with increased $\lambda$. Additionally, the result indicates as the $\lambda$ increases, subject fidelity can be slightly decreased. We hypothesize that this trade-off arises due to the model prioritizing textual context alignment over subject preservation at higher $\lambda$ values, as the MLM objective encourages the model to focus more on capturing the semantic relationships of the contexts.

**Impact of Contextual Semantics in Customizing Multi-concept Images.** We study whether the proposed method effectively guides the personal concept token to utilize the contextual semantics, by applying our method to learn a *single* concept from images containing *multiple* concepts (Figure 6). For this, we construct a set of 50 prompts that contextual semantics of the concept is highly

specific to the concept (e.g., "a man petting a [v] dog" for a dog). Surprisingly, applying our approach leads to successful learning of the targeted concept within the multiple concepts, leading to disentanglement results. This result indicates that, by training the concept embedding to predict the word that best aligns with its contexts, the concept embedding is driven to be semantically aligned with the context. As the overall semantics of the prompt set are highly specific to the concept (e.g., a dog), the concept embedding converges toward representing that specific concept (i.e., a dog).

**Impact of Masking Probability.** We train the model with different masking probabilities and study its impact. Table 4 shows that the performance is relatively insensitive to the masking probability, however, excessively high values lead to degradation. This result validates the importance of contextual information, as excessively high masking value leads to contextual semantics removal.

## 5.4 ADDITIONAL QUALITATIVE RESULTS

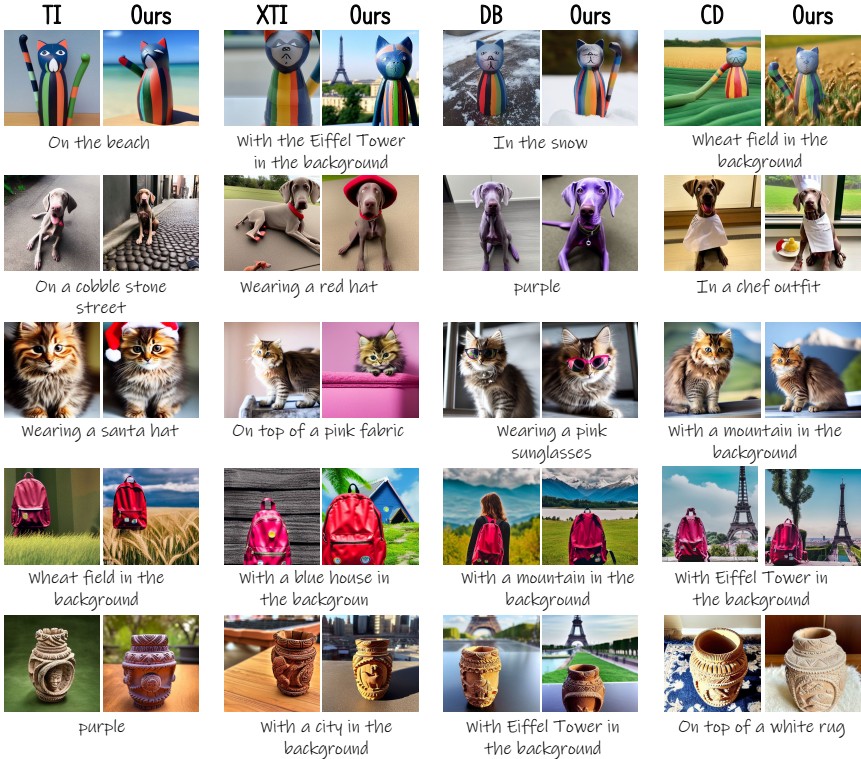

Figure 7: Additional qualitative comparison of the proposed method with the baselines. Our method is highly compatible with different methods. In general, compared to baseline approaches, the generated images from our approach show higher semantic alignment with the input prompt.

We present the additional qualitative comparison results of the proposed method in (Figure 7). In general, the baseline method that integrates our approach leads to improvement in prompt fidelity. In this visualization results, the baseline approach shows a higher tendency to neglect the context semantics. We analyze that the baseline approach loses the semantics of the context in textual space, which leads to the loss of semantics of the generated images. In contrast, the adoption of MLM leads to the preservation of the contextual semantics in text embedding, resulting in images with enhanced semantics with higher prompt fidelity. This visualization results further support our claim. We provide additional qualitative results of the proposed method in Appendix Section A.6.

## 6 CONCLUSION

In this paper, we proposed a highly cost-effective text-to-image customization method that enhances the semantics of the textual representation, thereby improving the semantic quality and prompt fidelity of the generated images. Our analysis revealed that the context overfitting problem in existing approaches stems from fine-tuning with limited contexts. We addressed this issue by diversifying the context of the personal concept solely within the textual space. By integrating our approach with different text-to-image customization methods, we observed consistent improvement in CLIP scores. The effectiveness of the proposed method is demonstrated through both theoretical analysis and extensive experimental validation.

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
