# A APPENDIX

## A.1 IMPLEMENTATION DETAILS OF BASELINES

**Textual Inversion.** For this baseline method, we train the model for 2,000 iterations with a constant learning rate `2e-3`. We use the batch size of 4 to train the method. Other than the concept token embeddings, no parameters are updated during the training. We set the batch size for MLM as 25.

**XTI.** For this baseline method, we train the model for 1,500 iterations with a constant learning rate of `2e-3`. We use the batch size of 4 to train the method. Following the original method, a set of multiple concept embeddings is utilized to be aligned with the same concept image. We set the batch size for MLM as 12 due to the memory limit.

**DreamBooth.** For this baseline method, we train the model for 1,000 iterations with a constant learning rate of `1e-6`. We use the batch size of 2 to train the method. Following the original method, the prior preservation loss is adopted during the training. For this, we generate a set of 200 images by prompting with "a picture of [SUBJECT CLASS]", by denoting the general class of the concept in the prompt. We update the parameters of both the CLIP text encoder and the diffusion U-Net. We set the batch size for MLM as 25.

**CustomDiffusion.** For this baseline method, we train the model for 5,00 iterations with a constant learning rate of `4e-5`. We use the batch size of 4 to train the method. Following the original method, we adopt prior preservation with generated images. During training only the Key/Value projection layers of the diffusion U-Net are updated during training. We set the batch size for MLM as 25.

## A.2 DETAILS OF TEXT PROMPT SET CONSTRUCTION

To generate a contextually diverse prompt set with minimal human intervention, we utilize a large pretrained language model (LLM) OpenAI (2023). Based on whether the personal concept is classified as living or nonliving, we predefined context categories and query the LLM to generate relevant elements for each category. The predefined categories for the living personal concepts are as below,

1. **Human Interactive Prompts:** A set of prompts that involves diverse interaction between different human subjects (e.g., *"Albert Einstein is watching TV with [V]"*).

2. **Relative Position Prompts:** A set of prompts that involves different positioning words and different objects (e.g., *"a picture of [V] next to a red vase"*).

3. **Background Prompts:** A set of prompts that describes a scene with different backgrounds (e.g., *"a picture of [V] with Eiffel Tower in the background"*).

4. **Image Style Prompts:** A set of prompts that describe image style (e.g., *"a picture of [V] in Pop Art style"*).

5. **Attributes Changing Prompts:** A set of prompts that describe the target concept with different visual attributes (e.g., *"a picture of [V] in blue sailor outfit"*).

Similarly, for non-living objects, we construct a set of prompts in five different types of contexts,

1. **Human Interactive Prompts:** A set of prompts that involves diverse interaction between different human subjects (e.g., *"Albert Einstein is watching TV with [V]"*).

2. **Relative Position Prompts:** A set of prompts that involves different positioning words and different objects (e.g., *"a picture of [V] next to a red vase"*).

3. **Background Prompts:** A set of prompts that describes a scene with different backgrounds (e.g., *"a picture of [V] with Eiffel Tower in the background"*).

4. **Image Style Prompts:** A set of prompts that describe image style (e.g., *"a picture of [V] in Pop Art style"*).

5. **Attributes Changing Prompts:** A set of prompts that describe the target concept with different visual attributes (e.g., *"a picture of [V] in blue sailor outfit"*).

### A.3 IMPLEMENTATION DETAILS OF CONTEXTUALIZER

Contextualizer constitutes four blocks of a self-attention layer and a feed-forward layer, followed by a layer normalization layer, where each block learns the residuals of the input with the residual connection. To train the contextualizer, we use a merged set of COCO caption dataset Chen et al. (2015) and the prompt set that we constructed. For the manual prompt set, we replace the personal concept token with the personal concept token to corresponding prior concept token. During training we set the ratio of batch of the two prompt set to be 70 to 30. The contextualizer is pretrained for 100K iterations with a learning rate of $1e-4$, and batch size 150. We use the AdamW optimizer Loshchilov (2017).

### A.4 JUSTIFICATION - SEMANTIC ENHANCEMENT IN TEXTUAL SPACE

The proof of Proposition 1.

*Proof.* Given an attention map $\mathbf{A}$, with $\sum_j \mathbf{A}[i,j] = 1$, $\mathbf{A}[i,j] \geq 0$, and the value matrix $\mathbf{V}$, the output of the attention layer is,

$$c_i = \sum_{j=1}^{N} \mathbf{A}[i,j]\mathbf{V}[j,:]. \tag{1}$$

The concept token at index $j_*$ has the highest attention value, *i.e.*, $\mathbf{A}[i,j_*] \gg \mathbf{A}[i,j], \forall j \neq j_*$. We have,

$$c_i = \sum_{j=1}^{N} \mathbf{A}[i,j]\mathbf{V}[j,:] = \sum_{j=1,j\neq*}^{N} \mathbf{A}[i,j]\mathbf{V}[j,:] + \mathbf{A}[i,j_*]\mathbf{V}[j_*,:] \approx \mathbf{A}[i,j_*]\mathbf{V}[j_*,:] \approx \mathbf{V}[j_*,:]. \tag{2}$$

The L2 norm between the text embeddings of the concept token $c_{i_*}$ and context tokens $c_i$ is,

$$\|c_i - c_{i_*}\|_2$$

$$= \|\sum_{j=1,j\neq j_*}^{N} (\mathbf{A}[i,j] - \mathbf{A}[i_*,j])\mathbf{V}[j,:] + (\mathbf{A}[i,j_*] - \mathbf{A}[i_*,j_*])\mathbf{V}[j_*,:]\|_2$$

$$\leq \|\sum_{j=1,j\neq j_*}^{N} (\mathbf{A}[i,j] - \mathbf{A}[i_*,j])\mathbf{V}[j,:]\|_2 + \|(\mathbf{A}[i,j_*] - \mathbf{A}[i_*,j_*])\mathbf{V}[j_*,:]\|_2$$

$$\leq \sum_{j=1,j\neq j_*}^{N} \|\mathbf{A}[i,j] - \mathbf{A}[i_*,j]\|_2\|\mathbf{V}[j,:]\|_2 + \|\mathbf{A}[i,j_*] - \mathbf{A}[i_*,j_*]\|_2\|\mathbf{V}[j_*,:]\|_2. \tag{3}$$

Suppose $\mathbf{A}[i,j_*] = 1 - \delta_{ij_*}$ and $\mathbf{A}[i_*,j_*] = 1 - \delta_{i_*j_*}$, where $\mathbf{0} \leq \delta_{ij_*} < \delta$ and $\mathbf{0} \leq \delta_{i_*j_*} < \delta$. $\mathbf{A}[i,j] = \delta_{ij}, \forall j \neq j_*$, $\mathbf{A}[i_*,j] = \delta_{i_*j}, \forall j \neq j_*$, $0 \leq \delta_{ij} < \delta$ and $0 \leq \delta_{i_*j} < \delta$, where $\delta$ is a small value. We have $\|\delta_{ij} - \delta_{i_*j}\|_2 < \delta$ and $\|\delta_{i_*j_*} - \delta_{ij_*}\|_2 < \delta$. Thus,

$$\|c_i - c_*\|_2$$

$$\leq \sum_{j=1,j\neq j_*}^{N} \|\delta_{ij} - \delta_{i_*j}\|_2\|\mathbf{V}[j,:]\|_2 + \|\delta_{i_*j_*} - \delta_{ij_*}\|_2\|\mathbf{V}[j_*,:]\|_2$$

$$\leq \delta \sum_{j=1,j\neq j_*}^{N} \|\mathbf{V}[j,:]\|_2 + \delta\|\mathbf{V}[j_*,:]\|_2. \tag{4}$$

Since $\|\mathbf{V}[j,:]\|_2$ is bounded, we have,

$$\|c_i - c_*\|_2 \leq \delta_{\mathbf{V}}, \tag{5}$$

where $\delta_{\mathbf{V}} = \delta \sum_{j=1}^{N} \|\mathbf{V}[j,:]\|_2$. $\qquad\square$

The proof of Proposition 2.

*Proof.* Suppose $\|c_b - \hat{c}_b\|_2$ is a small value, using the Taylor series, we have,

$$\mathcal{L}_{\text{MLM}}(c_b) = \mathcal{L}_{\text{MLM}}(\hat{c}_b) + (c_b - \hat{c}_b)^T \text{grad}(\mathcal{L}_{\text{MLM}}(\hat{c}_b)) + \mathcal{O}(c_b - \hat{c}_b)$$
$$\approx \mathcal{L}_{\text{MLM}}(\hat{c}_b) + (c_b - \hat{c}_b)^T \text{grad}(\mathcal{L}_{\text{MLM}}(\hat{c}_b)), \tag{6}$$

where $\text{grad}(\cdot)$ is the first-order derivative. Using Cauchy-Schwartz inequality, we have,

$$(c_b - \hat{c}_b)^T \text{grad}(\mathcal{L}_{\text{MLM}}(\hat{c}_b)) \leq \|c_b - \hat{c}_b\|_2 \cdot \|\text{grad}(\mathcal{L}_{\text{MLM}}(\hat{c}_b))\|_2. \tag{7}$$

Since $\hat{c}_b$ is near the optimal value, which is achieved by optimizing the contextualizer, we have $\text{grad}(\mathcal{L}_{\text{MLM}}(\hat{c}_b)) \leq \delta_g$, where $\delta_g$ is a small value. Therefore, we have

$$\mathcal{L}_{\text{MLM}}(c_b) - \mathcal{L}_{\text{MLM}}(\hat{c}_b) \leq \delta_g \|c_b - \hat{c}_b\|_2. \tag{8}$$

$\square$

### A.5 JUSTIFICATION - SEMANTIC ENHANCEMENT IN IMAGE SPACE

The proof of Proposition 4.

*Proof.* The image embedding $\mathbf{z}$ and text embedding $\mathbf{C}$ are projected as $Q_{\mathcal{I}} = \mathbf{z}\mathbf{W}_Q, K_{\mathcal{T}} = \mathbf{C}\mathbf{W}_K$. For text embeddings at indices $i$ and $j$, we have,

$$\mathbf{K}_{\mathcal{T}}[i,:] = c_i \mathbf{W}_K \tag{9}$$
$$\mathbf{K}_{\mathcal{T}}[j,:] = c_j \mathbf{W}_K. \tag{10}$$

The relation map is $\mathbf{M} = \mathbf{Q}_{\mathcal{I}} K_{\mathcal{T}}^T$, and $\mathbf{M}[:,i] = \mathbf{Q}_{\mathcal{I}} K_{\mathcal{T}}[i,:], \mathbf{M}[:,j] = \mathbf{Q}_{\mathcal{I}} K_{\mathcal{T}}[j,:]$. Thus,

$$\|\mathbf{M}[:,i] - \mathbf{M}[:,j]\|_2 = \|\mathbf{Q}_{\mathcal{I}}(K_{\mathcal{T}}[i,:] - K_{\mathcal{T}}[j,:])\|_2$$
$$\leq \|\mathbf{Q}_{\mathcal{I}}\|_F \|(K_{\mathcal{T}}[i,:] - K_{\mathcal{T}}[j,:])\|_2$$
$$= \|\mathbf{Q}_{\mathcal{I}}\|_F \|(c_i - c_j)\mathbf{W}_K\|_2$$
$$\leq \|\mathbf{Q}_{\mathcal{I}}\|_F \|\mathbf{W}_K\|_F \|(c_i - c_j)\|_2$$
$$= \alpha \|(c_i - c_j)\|_2, \tag{11}$$

where $\|\cdot\|_F$ is Frobenius norm, and $\alpha = \|\mathbf{Q}_{\mathcal{I}}\|_F \|\mathbf{W}_K\|_F$. $\square$

### A.6 ADDITIONAL QUALITATIVE EXAMPLES

We provide additional qualitative results of our approach combined with each baseline method, TI (Gal et al., 2022), XTI(Voynov et al., 2023), DB(Ruiz et al., 2023) and CD(Kumari et al., 2023). We provide two types of generation results, living or non-living objects (Figures 1, 2, 3, 4, 5, 6, 7,8)

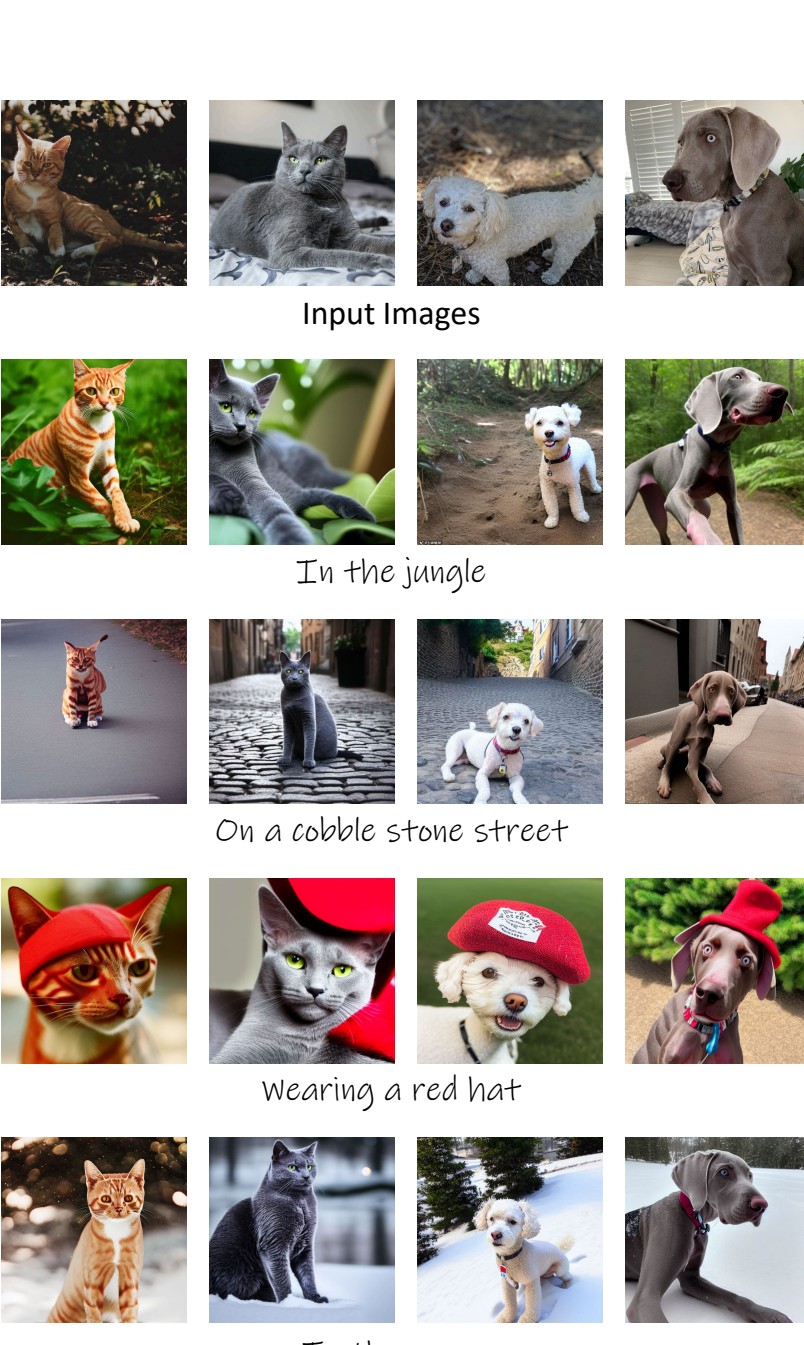

Input Images

In the jungle

On a cobble stone street

Wearing a red hat

In the snow

Figure 1: Additional Qualitative Result of TI - Living Objects.

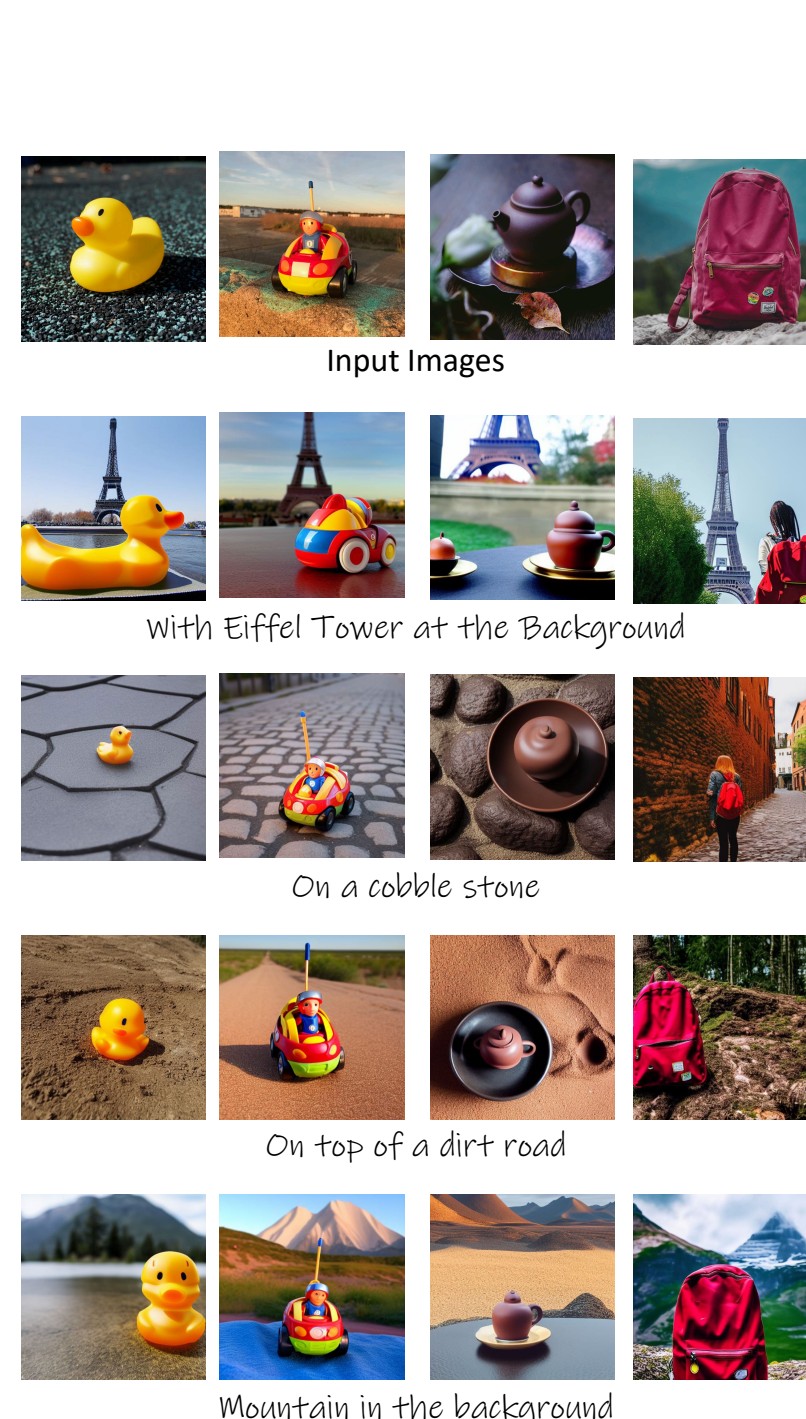

Input Images

With Eiffel Tower at the Background

On a cobble stone

On top of a dirt road

Mountain in the background

Figure 2: Additional Qualitative Result of TI - Non-living Objects.

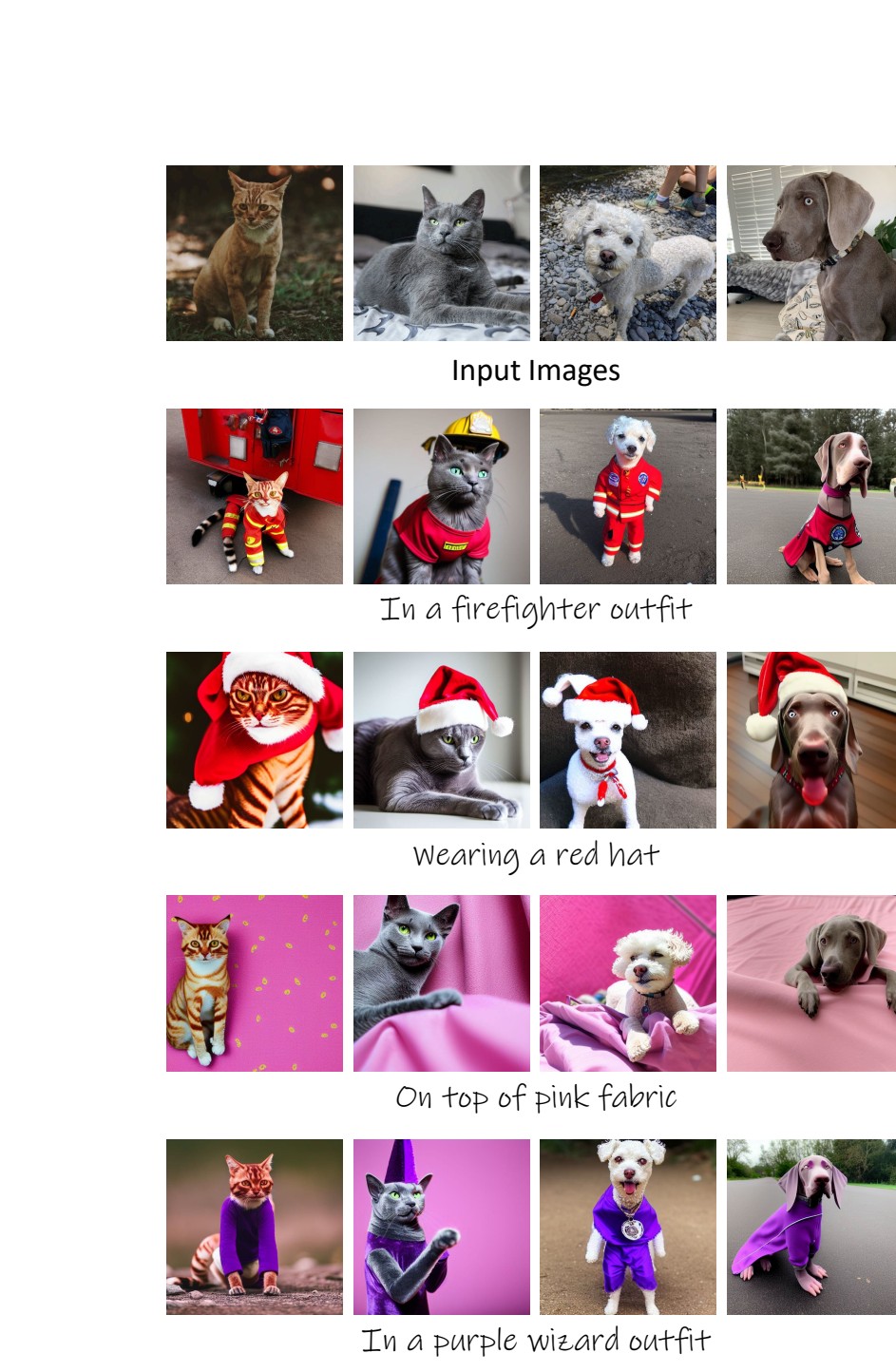

Input Images

In a firefighter outfit

Wearing a red hat

On top of pink fabric

In a purple wizard outfit

Figure 3: Additional Qualitative Result of XTI - Living Objects.

1026
1027
1028
1029
1030
1031
1032
1033
1034
1035
1036
1037
1038
1039
1040
1041
1042
1043
1044
1045
1046
1047
1048
1049
1050
1051
1052
1053
1054
1055
1056
1057
1058
1059
1060
1061
1062
1063
1064
1065
1066
1067
1068
1069
1070
1071
1072
1073
1074
1075
1076
1077
1078
1079

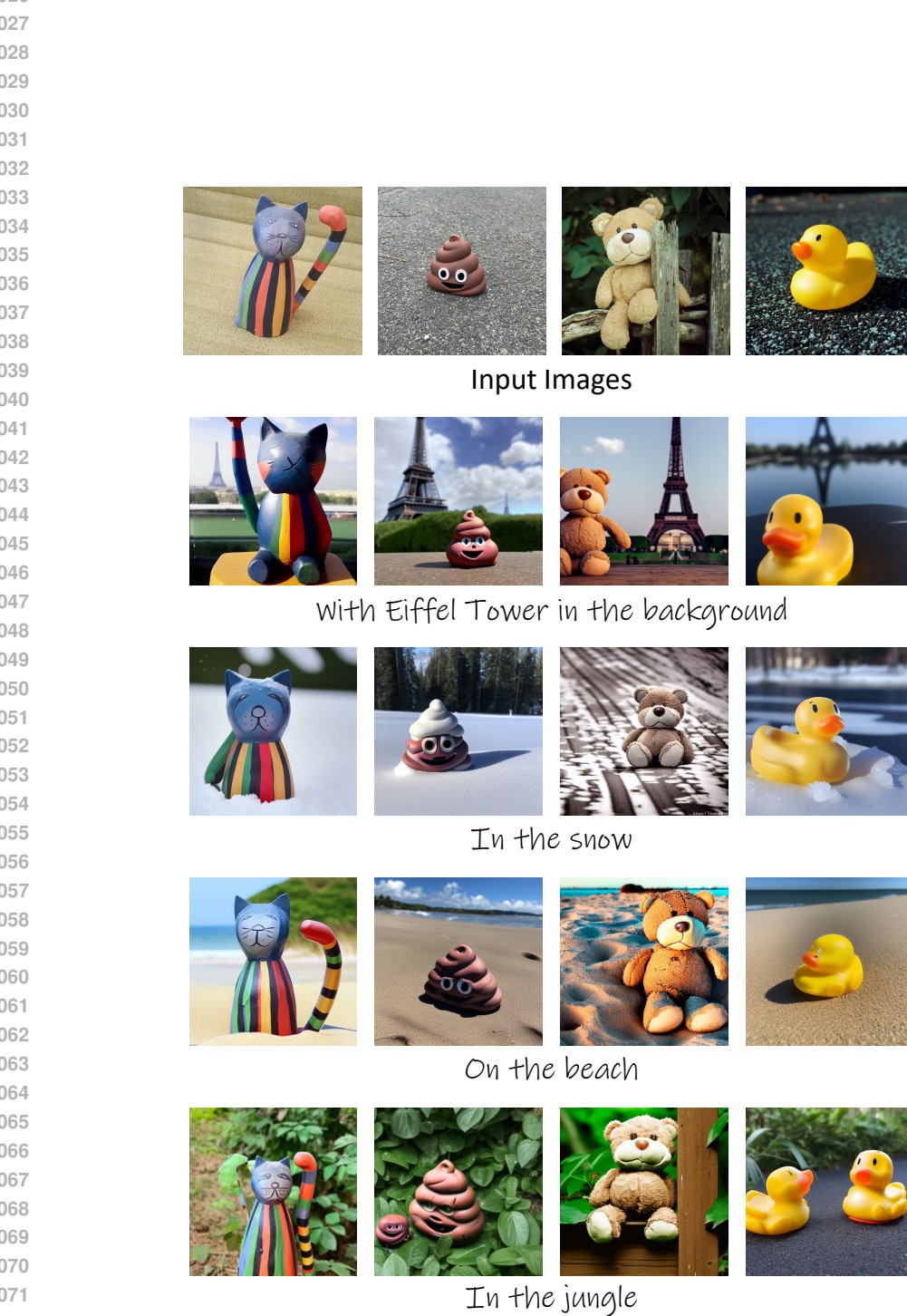

Figure 4: Additional Qualitative Result of XTI - Non-living Objects.

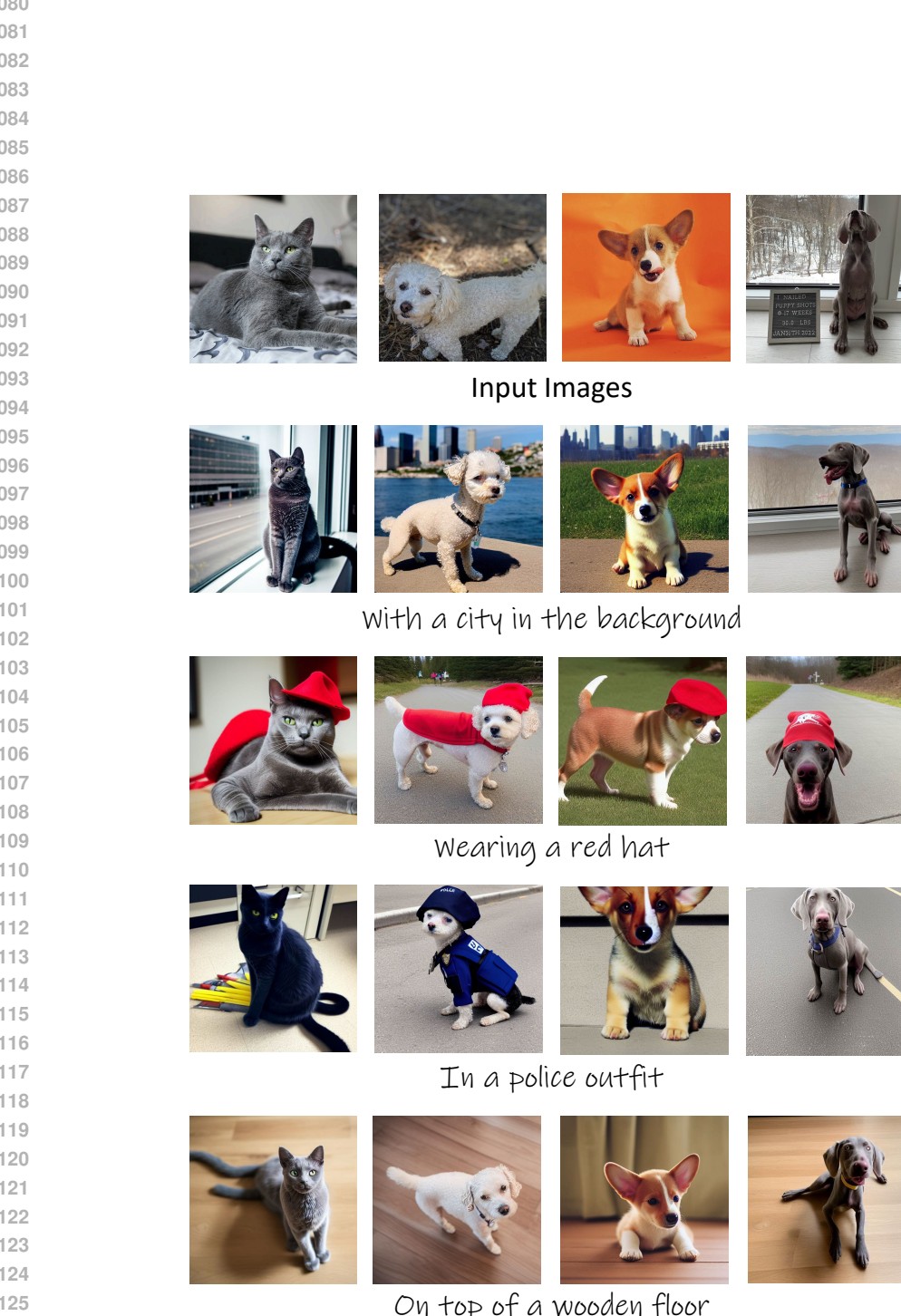

Input Images

With a city in the background

Wearing a red hat

In a police outfit

On top of a wooden floor

Figure 5: Additional Qualitative Result of DB - Living Objects.

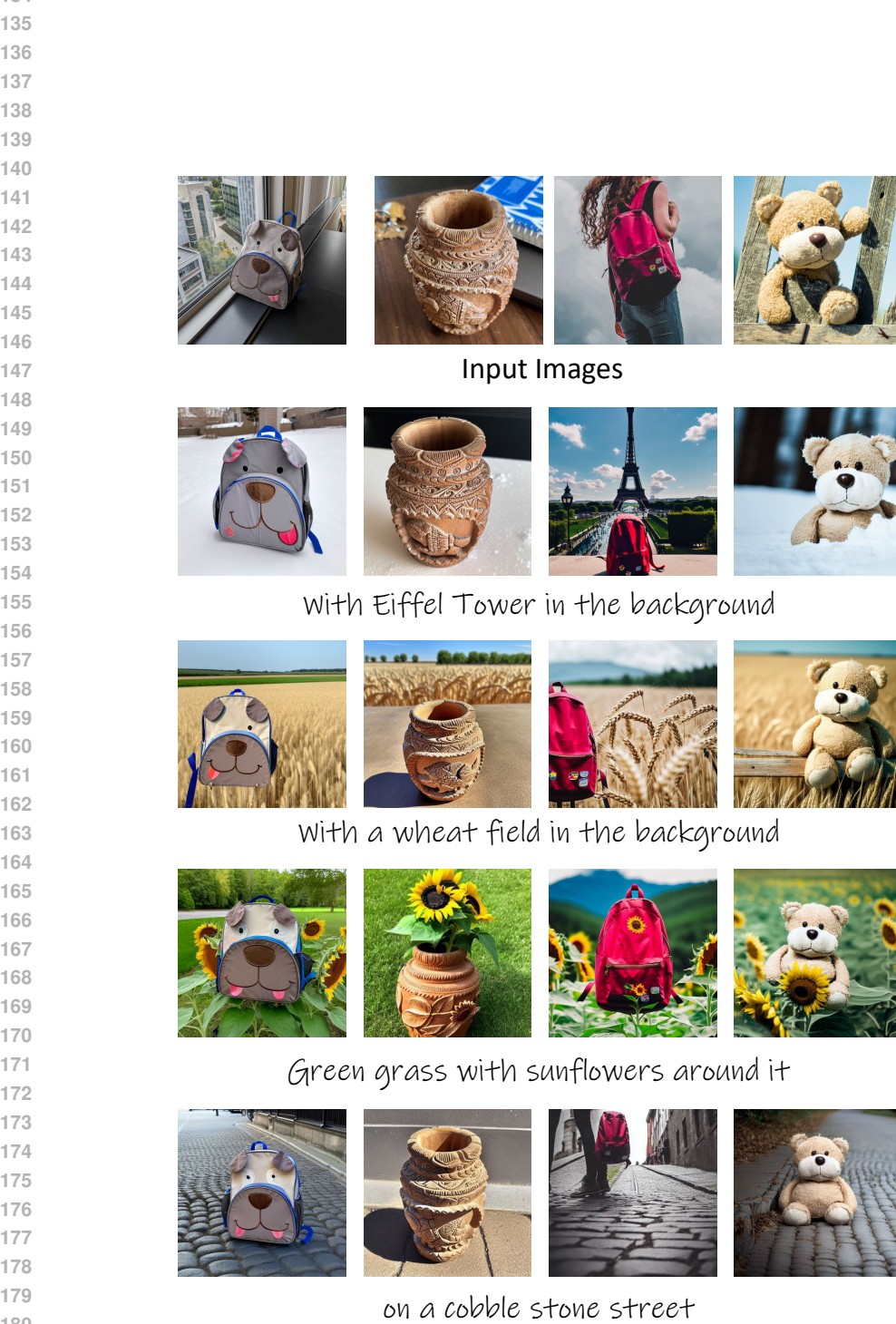

Input Images

With Eiffel Tower in the background

With a wheat field in the background

Green grass with sunflowers around it

on a cobble stone street

Figure 6: Additional Qualitative Result of DB - Non-living Objects.

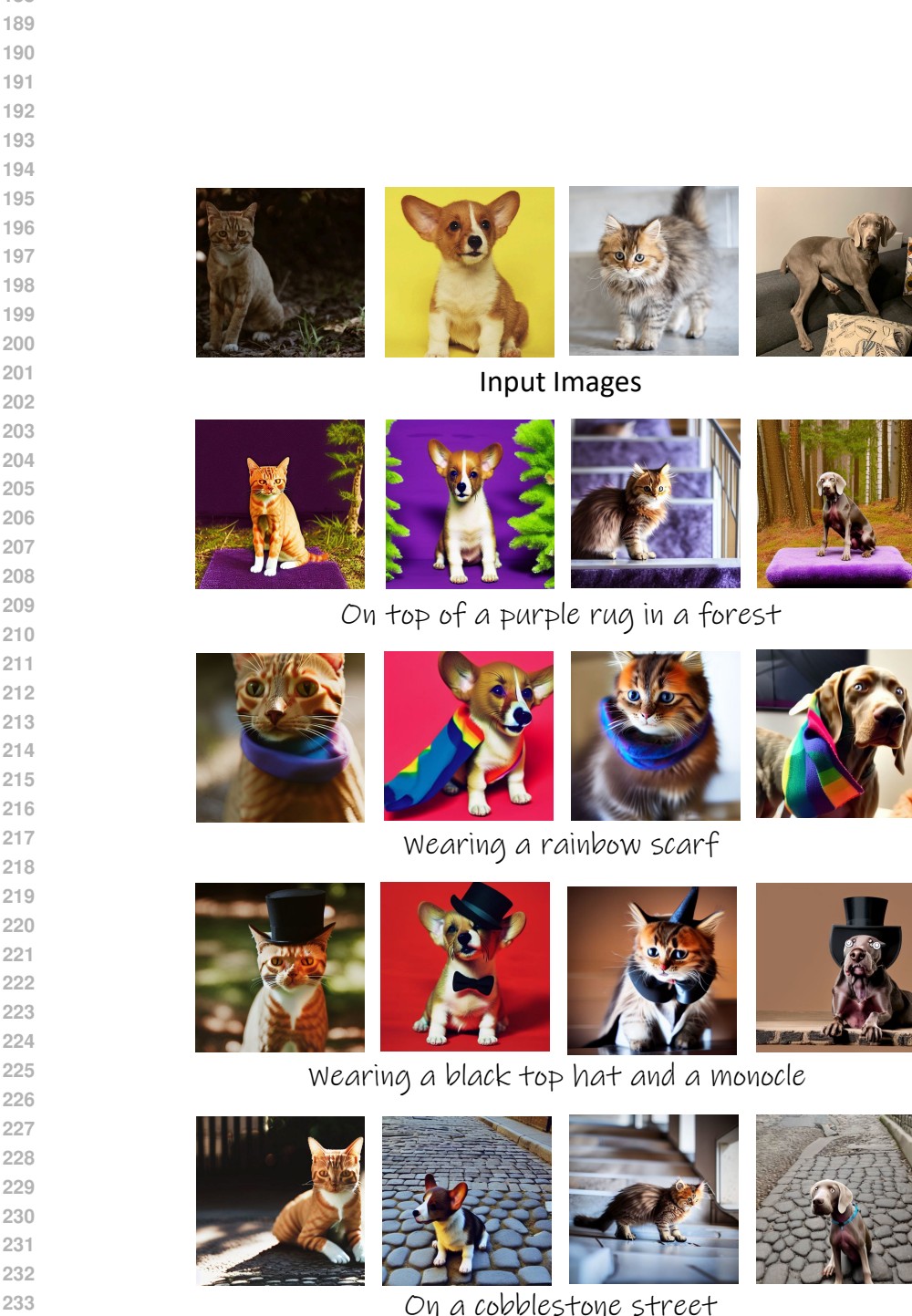

Input Images

On top of a purple rug in a forest

Wearing a rainbow scarf

Wearing a black top hat and a monocle

On a cobblestone street

Figure 7: Additional Qualitative Result of CD - Living Objects.

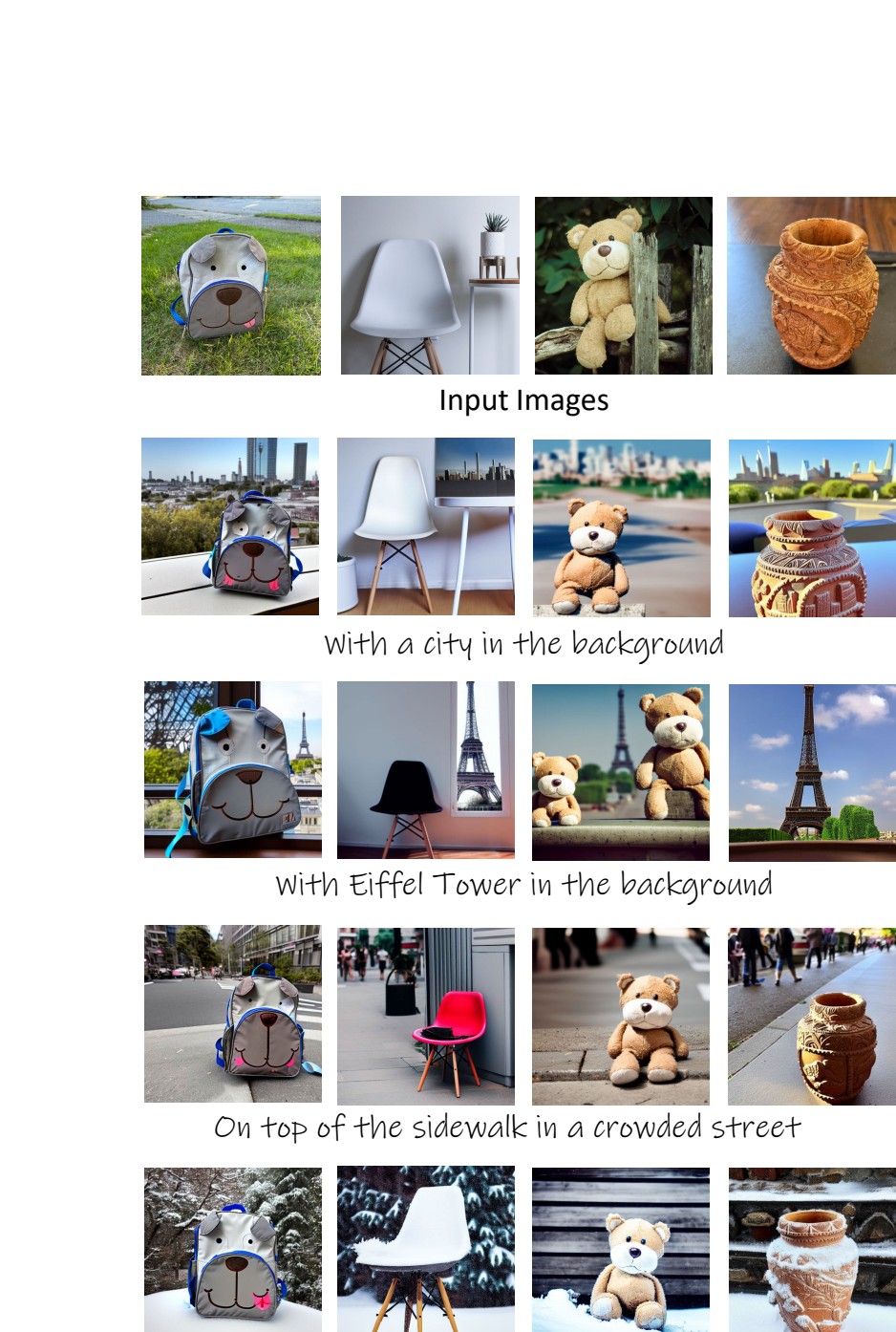

Input Images

With a city in the background

With Eiffel Tower in the background

On top of the sidewalk in a crowded street

In the snow

Figure 8: Additional Qualitative Result of CD - Non-living Objects.