# OpenReview forum: "Learning to Customize Text-to-Image Diffusion In Diverse Context"
_ICLR.cc/2025/Conference — ICLR 2025 Conference Withdrawn Submission_

### Official Review · Reviewer_LzGn · 2024-11-02

**Soundness:** 3
**Presentation:** 3
**Contribution:** 2
**Rating:** 5
**Confidence:** 3

**Summary:**

This paper proposes a method to mitigate concept overfitting in diffusion-based text-to-image models by diversifying the context of personal concepts through diverse text prompts and a Masked Language Modeling objective, leading to improved alignment between text and generated images.

**Strengths:**

1. The proposed method was shown to be robust, and the performance is competitive, showing the effectiveness of the designs.

2. The writing of this paper is easy to follow.

**Weaknesses:**

1. The proposed method does not outperform DB and CD in terms of DINO and DINO-FG metrics, which weakens the evidence for its effectiveness.

2. The proposed method may introduce some additional image concepts, such as (Row 4, CD vs. Ours) and (Row 5, DB vs. Ours), where new visual elements (e.g., leaves or tree) appear in the images generated by the proposed method. This diminishes the effectiveness of the proposed method.

**Questions:**

see the weakness.

---

### Official Review · Reviewer_hAgW · 2024-11-03

**Soundness:** 2
**Presentation:** 2
**Contribution:** 2
**Rating:** 5
**Confidence:** 5

**Summary:**

The manuscript introduces propose the strategies to resolve the limitation of overfitting to the small subset of training images and hard to generalize to new contexts with different text prompt input. In practice, the method aims to create a rich set of text prompt and the use of unsupervised learning objective to enhance the ability of concept personalization with the new text prompt input. The results demonstrate that the method is competitive with leading frameworks in various image generation tasks.

**Strengths:**

The authors have developed a framework for generating personalized images that effectively integrates the context diversification of personal concept using masked language modeling and to solve the issue of concept overfitting.

The paper provides experimental results, including both quantitative and qualitative assessments, showcasing the superior performance of the framework. The results clearly highlight the effectiveness of the proposed method in facilitating personalized image generation.

**Weaknesses:**

In Figure 3, the method utilizes Masked Language Modeling module to enhance the identity details. However, it is unclear how it would perform with fine-grained subjects (two dogs or two cats with different breed). Also, I wonder how this module would work when the concept size increases. Clarification is needed on whether the module can effectively manage such fine distinctions and multiple diverse subjects.

Recent methodologies [1, 2] have demonstrated the capability to learn multi-concept personalization, it remains uncertain if the proposed work can handle multiple personalized instances (> 2), particularly for contexts involving up to five subjects. The absence of qualitative results for three or more subjects in both the main text and appendix might be a notable omission. Including these results would substantiate the method's capability in more complex scenarios.

[1] Liu, Zhiheng, et al. "Cones 2: Customizable image synthesis with multiple subjects." arXiv preprint arXiv:2305.19327 (2023).

[2] Yeh, Chun-Hsiao, et al. "Gen4Gen: Generative Data Pipeline for Generative Multi-Concept Composition." arXiv preprint arXiv:2402.15504 (2024).

**Questions:**

Given the concerns mentioned, particularly around the method's scalability to more complex multi-subject personalizations and the clarification behind the Masked Language Modeling module, I recommend a "marginally below the acceptance threshold" for this paper. Enhancements in demonstrating multi-subject capabilities, clarity in embedding visualization, and justification for the choice of technology could potentially elevate the manuscript to meet publication standards.

---

### Official Review · Reviewer_HSH9 · 2024-11-04

**Soundness:** 1
**Presentation:** 3
**Contribution:** 2
**Rating:** 3
**Confidence:** 4

**Summary:**

This paper proposes a method for customization of text-to-image (T2I) generation. The authors propose using the masked language modeling (MLM) objective on the sequences of texts combined with concept tokens, where the text encoder is required to predict the partially masked context.

**Strengths:**

- In terms of originality, although masked language modeling (MLM) is nothing new, it is a bit novel to use it in the customization of T2I generation.
- The writing of this paper is generally clear. The method is described clearly and is easy to understand, although a few texts are unclear (see weaknesses).
- The proposed method enhances the alignment between the generated images and the text prompts. From the qualitative results, I can see that the method results in better alignment. The quantitative metric CLIP-T is also shown to be improved for the proposed method.

**Weaknesses:**

- The subject fidelity metrics (DINO and DINO-FG) of the proposed method are worse than the baselines, including DreamBooth (DB) and CustomDiffusion (CD). It shows that the proposed method may increase text-image alignment CLIP-T at the cost of reduced subject fidelity.
- I found the theories unconvincing. The authors want to use Proposition 1 to prove the distance between the context tokens and concept tokens is bounded by a small value $\delta_V$. Firstly, I do not understand why the value matrix norm $\|V[j, :]\|$ is said to be bounded (line 804 in the appendix). Secondly, I cannot see why $\delta_V$ is said to be small. How small is $\delta_V$? Because of these questions, I cannot understand the benefits of using Masked Language Modeling in the customization of T2I generation.
- A minor point regarding the organization of the paper is that there are some unclear parts in the texts. For example, the meaning of $c_i$ is not mentioned in Proposition 1. Although I could find its meaning in Remark 5 later, it would be better if it is explained in the first place it appears.

**Questions:**

See weaknesses.

---

### Official Review · Reviewer_qWhR · 2024-11-07

**Soundness:** 2
**Presentation:** 3
**Contribution:** 2
**Rating:** 5
**Confidence:** 3

**Summary:**

This paper presents a customization approach for text-to-image diffusion models that aims to improve prompt fidelity by diversifying the textual contexts in which concept tokens are used. The proposed method constructs a set of contextually diverse prompts, uses a Masked Language Modeling (MLM) objective, and claims to enhance the generalization and prompt fidelity of customized diffusion models. The authors demonstrate the applicability of this approach on multiple baseline methods, including Textual Inversion and DreamBooth, with results showing improvements in prompt fidelity metrics.

**Strengths:**

- The paper is generally well-organized and provides clear explanations of the method, including detailed descriptions of the MLM application.
- The proposed approach requires no architectural modifications and the proposed approach avoids generating new image pairs, reducing computational costs.

**Weaknesses:**

The main weakness is that the experimental evaluation is not convincing.
- Missing evaluation of the quality of the generated personal concept: image similarity between the generated personal concept and the reference personal concept has to be evaluated.
- The selected personal concepts are fairly weak in the sense that most concepts do not have “unique” personal chrematistics. Instead, most of them have been learned during SD model pretraining already. More unique personal concepts should be evaluated.
- Missing comparison with baseline methods: for example, Key-Locked Rank One Editing for Text-to-Image Personalization, SIGGRAPH 2023.

**Questions:**

Please refer to the weakness section.

**Details Of Ethics Concerns:**

It is an arxiv paper at

https://arxiv.org/abs/2410.10058

---

### Note · Authors · 2024-11-12

**Comment:**

I have read and agree with the venue's withdrawal policy on behalf of myself and my co-authors.

**Withdrawal Confirmation:**

I have read and agree with the venue's withdrawal policy on behalf of myself and my co-authors.